# Naturally-Coupled Dark Sectors

**Durmuş Demir** 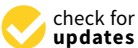

Faculty of Engineering and Natural Sciences, Sabancı University, Tuzla, İstanbul 34956, Turkey; durmus.demir@sabanciuniv.edu

**Abstract:** The dark sector, composed of fields that are neutral under the standard model (SM) gauge group, can couple to the SM through the Higgs, hypercharge and neutrino portals, and pull the SM towards its scale by loop corrections. This instability, which is not possible to prevent in the known SM completions, such as supersymmetry, due to the sizable couplings to the SM, calls for alternative mechanisms that can neutralize sensitivities of the SM to the dark sector scale and to the ultraviolet cutoff above it. Here we review such a mechanism in which incorporation of gravity into the SM predicts the existence of a dark sector and allows it to be naturally coupled to the SM. We discuss and illustrate salient processes that can probe the naturally coupled dark sectors.

**Keywords:** hierarchy problem; naturally coupled dark sector; emergent gravity

## 1. Introduction

The standard model of elementary particles (SM), spectrally completed with the discovery of the Higgs boson at the LHC [1], has so far shown excellent agreement with all the experimental data [2]. It is a renormalizable quantum field theory (QFT). Possible new physics beyond the SM (BSM), whose scale puts an ultraviolet (UV) momentum cutoff $\Lambda$ on the SM loops, must step in at $\Lambda \simeq$ TeV if the SM Higgs boson mass is to remain at its experimental value [1,3–5]; however, the LHC experiments have shown that the SM continues to hold well [6] up to energies well above its TeV cutoff.

This reign of the SM, an experimental fact, contradicts the existence of various phenomena begging for a consistent explanation. Indeed, there are astrophysical (dark matter, dark photon), cosmological (dark energy, inflation) and other (neutrino masses, flavor, unification, $\cdots$) phenomena that cannot be accounted for within the SM; therefore, we need to have a BSM sector around and beyond a TeV [7,8]. In general, the SM + BSM is a coverall renormalizable QFT, with possible non-renormalizable interactions induced by high-scale physics.

The SM-singlet BSM fields, which form dark energy, dark matter and possibly more, make up the dark sector (DS). The DS, which forms 95% of the Universe [9–11], is known via only its gravitational interactions. The efforts to detect it by experiments [12–17] and observations [18–22] are continuing. The DS and SM can have renormalizable and non-renormalizable interactions. At the renormalizable level, they couple via three distinct portals: The Higgs portal [23]

$$\lambda^2_{H'_i}(H^\dagger H)(H'^\dagger_i H'_i) \tag{1}$$

through which the DS scalars $H'_i$ couple to the SM Higgs doublet $H$, the hypercharge portal [24]

$$\lambda_{V'_i} V'_{i\mu\nu} B^{\mu\nu} \tag{2}$$

through which the DS gauge bosons $V_i'^\mu$ couple to the SM hypercharge gauge boson $B_\mu$, and finally the neutrino portal

$$\lambda_{N_i'}\overline{L}HN_i' + \text{h.c.} \tag{3}$$

through which the DS right-handed neutrinos $N_i'$ couple to the SM Higgs doublet and the lepton doublet $L$.

The SM-nonsinglet BSM fields, which are charged under the SM gauge group, make up the visible sector (VS). The VS interacts with the SM via the usual gauge interactions and via the Higgs portal in (1) and the hypercharge portal in (2). It is severely constrained by collider bounds [2,6].

The fundamental problem with the QFTs is that loop corrections throw their bosonic sectors at the UV boundary. The Higgs boson mass $m_h$, for instance, acquires a loop correction ($\lambda^2_{SM-BSM} = \lambda^2_{H',V',N'}$ where $X'$ can be both DS and VS fields)

$$\delta m_h^2 = c_h\Lambda_\wp^2 + \sum_F c_{hBSM}\lambda^2_{SM-BSM}M^2_{BSM}\log\frac{M^2_{BSM}}{\Lambda_\wp^2} \tag{4}$$

which removes the SM from its natural scale (the Fermi scale) when the UV momentum cutoff $\Lambda_\wp$ is large ($\Lambda_\wp \gg m_h$) or when the BSM scale is large ($M_{BSM} \gg m_h$), with few percent loop factors $c_h$ and $c_{hBSM}$. It is clear that the mechanism that neutralizes the $\Lambda_\wp^2$ contribution in (4) must also keep the heavy BSM contribution small by *allowing* the SM-BSM couplings $\lambda_{SM-BSM}$ to be sufficiently small. This actually is a crucial point because the known completions of the SM (supersymmetry, extra dimensions, compositeness and their hybrids) have been sidelined by the LHC experiments due to the fact that their BSM sectors (superpartners in supersymmetry, Kaluza–Klein levels in extra dimensions and technifermions in compositeness) require $\lambda_{SM-BSM} \simeq \lambda_{SM}$ as a result of their defining symmetries. Thus, if the electroweak scale is to be stabilized with a view to satisfying the LHC bounds then there must exist a mechanism that

$$\text{neutralizes the } \Lambda_\wp^2 \text{ term in (4), and} \tag{5}$$

$$\text{allows for SM-BSM couplings } \lambda_{SM-BSM} \text{ to be sufficiently small} \tag{6}$$

where $\lambda_{SM}$ is a typical SM coupling. In search for a mechanism that can accomplish both (5) and (6), one realizes that the SM fails to cover two major physical phenomena:

$$\text{BSM sector (dark matter, inflaton, right-handed neutrinos, \dots ), and} \tag{7}$$

$$\text{gravity.} \tag{8}$$

This means that the mechanism that accomplishes (5) and (6) must account for both (7) and (8). In other words, it must be able to form a satisfactory BSM sector and incorporate gravity into SM + BSM.

In the present work we review a mechanism that predicts (7), covers (8) and accomplishes both (5) and (6). The mechanism is essentially an extension of Sakharov's induced gravity into gauge sector [25–27]. The review is based on the papers [28–32] and the talks [33–37].

In Section 2 below we review the aforementioned mechanism. We show that the incorporation of gravity into the SM predicts a BSM sector and allows SM-BSM couplings to be sufficiently small. This will constitute the naturally coupled dark sector emphasized in the title. (This section is largely based on [28,29,33,34].)

In Section 3 we give an in-depth discussion of the naturally coupled dark sector, and study certain processes characteristic of its salient features. We review there collider and

dark matter signatures along with a general classification of the dark sector. (This section is largely based on [28,29,38,39].)

In Section 4 we give future prospects and conclude.

## 2. Dark Sector: A Necessity for QFT–GR Concord

The SM needs be furthered for gathering a BSM sector and involving the gravity. This necessitates QFTs to be brought together with the gravity. In essence, what needs be done is to carry the QFTs into curved spacetime concordantly.

In an effort to realize the requisite concord, it proves useful to start with the classical field theory limit [40]. Then, taken as a classical field theory governed by an action $S_{cl}(\eta, \psi, \partial\psi)$ of the classical fields $\psi$ in the flat spacetime of metric $\eta_{\mu\nu}$, the SM can be carried into curved spacetime of a putative curved metric $g_{\mu\nu}$ by letting

$$S_{cl}(\eta, \psi, \partial\psi) \hookrightarrow S_{cl}(g, \psi, \nabla\psi) \tag{9}$$

in accordance with the equivalence principle map [41]

$$\eta_{\mu\nu} \hookrightarrow g_{\mu\nu}, \ \partial_\mu \hookrightarrow \nabla_\mu \tag{10}$$

in which $\nabla_\mu$ is the covariant derivative of the Levi–Civita connection

$${}^g\Gamma^\lambda_{\mu\nu} = \frac{1}{2} g^{\lambda\rho} \left( \partial_\mu g_{\nu\rho} + \partial_\nu g_{\rho\mu} - \partial_\rho g_{\mu\nu} \right) \tag{11}$$

so that $\nabla_\alpha g_{\mu\nu} = 0$ in parallel with $\partial_\alpha \eta_{\mu\nu} = 0$. The Levi–Civita connection determines the Ricci curvature $R_{\mu\nu}({}^g\Gamma)$ and the scalar curvature $R(g) = g^{\mu\nu} R_{\mu\nu}({}^g\Gamma)$.

The image of the SM in (9) is incomplete in regard to both (7) and (8). These sectors must added by hand since the equivalence principle map in (10) can induce neither the BSM sector nor the gravity (curvature sector). The BSM sector can be modeled by a classical field theory with generic couplings to the SM. The curvature sector, on the other hand, must have the form

$$\text{``curvature sector''} = \int d^4x \sqrt{-g} \left\{ -\frac{\tilde{M}^2}{2} R(g) + \tilde{c}_2 R(g)^2 + \frac{\tilde{c}_3}{\tilde{M}^2} R(g)^3 + \dots \right\} \tag{12}$$

if it is to lead to the general relativity (GR) [42–44]. Thus, it is with judiciously added BSM and curvature sectors that there arises a proper setup. To realize QFT–GR concord, all that is needed is the quantization of the curved spacetime classical action in (9) with the curvature sector in (12). This, however, is not possible simply because it is not possible to quantize the GR [45,46]. It is this non-renormalizable nature of GR that prevents reconciliation of the SM + BSM with the GR as two proper QFTs.

This non-quantizable nature of the GR obstructs the QFT–GR concord. To overcome this obstruction, it proves efficacious to take gravity classical as it is already not quantizable and adopt the effective SM as it resembles the classical SM in (9) in view of its long-wavelength field spectrum and loop-corrected couplings. To see where this approach leads to, it is first necessary to construct the SM effective action. The matter loops whose momenta vary in the range ($\Lambda_\wp$ explicitly breaks the Poincare invariance [47,48])

$$-\Lambda_\wp^2 \leq \eta^{\mu\nu} \ell_\mu \ell_\nu \leq \Lambda_\wp^2 \tag{13}$$

modify the classical action $S_{cl}(\eta, \psi)$ by adding an anomalous gauge boson mass term

$$\delta S_V(\eta, \Lambda_\wp) = \int d^4x \sqrt{-\eta}\, c_V \Lambda_\wp^2 \operatorname{Tr}\left[\eta_{\mu\nu} V^\mu V^\nu\right] \tag{14}$$

which breaks gauge symmetries explicitly and leads therefore to explicit color and charge breaking (CCB) [49,50]. The loop factor $c_V$ is given in Table 1 for the SM gauge bosons.

The matter loops add power-law corrections

$$\delta S_{\varnothing \phi}(\eta, \Lambda_{\wp}) = \int d^4x \sqrt{-\eta} \left\{ -c_{\varnothing} \Lambda_{\wp}^4 - \sum_i c_{\psi_i} m_i^2 \Lambda_{\wp}^2 - c_{\phi} \phi^2 \Lambda_{\wp}^2 \right\} \quad (15)$$

which push the scalar masses $m_{\phi}$ towards the UV scale to give cause to the big hierarchy problem [3–5] (the loop factor $c_{\phi}$ is given in Table 1 for the SM Higgs boson). They give rise to a UV-sized vacuum energy (the loop factors $c_{\varnothing}$ and $c_{\psi_i}$ are given in Table 1), with of course no physical consequences in the flat spacetime [51–54].

Finally, the matter loops also add logarithmic corrections

$$\delta S(\eta, \psi, \log \Lambda_{\wp}) \supset - \sum_i \hat{c}_{\psi_i} m_i^4 \log \frac{m_i^2}{\Lambda_{\wp}^2} - \sum_{i,\phi} \hat{c}_{\phi\psi_i} m_i^2 \log \frac{m_i^2}{\Lambda_{\wp}^2} \phi^2 - \sum_{i,f} \hat{c}_{f\psi_i} \log \frac{m_i^2}{\Lambda_{\wp}^2} m_f \bar{f} f \quad (16)$$

which respect all the symmetries of the QFT (as in the dimensional regularization [55–57]).

**Table 1.** The loop factors $c_V$ at one loop and various problems they cause (taken from Reference [33]).

| Loop Factor | SM Fields | SM Value | Problems Caused |
|---|---|---|---|
| $c_V$ | gluon | $\frac{21 g_3^2}{16\pi^2}$ | color breaking |
| $c_V$ | weak gauge bosons | $\frac{21 g_2^2}{16\pi^2}$ | isospin breaking |
| $c_V$ | hypercharge gauge boson | $\frac{39 g_1^2}{32\pi^2}$ | hypercharge breaking |
| $c_{\phi}$ | Higgs boson $h$ | $\frac{g_2^2 \text{str}[m^2]}{8\pi^2 M_W^2} \approx -\frac{g_2^2 m_i^2}{\pi^2 M_W^2}$ | big hierarchy problem |
| $c_{\varnothing} = -\frac{\text{str}[1]}{128\pi^2}$ | over all the SM fields | $\frac{31}{32\pi^2}$ | none (flat spacetime) |
| $\sum_i c_{\psi_i} m_i^2 = \frac{\text{str}[m^2]}{32\pi^2}$ | over all the SM fields | $\approx -\frac{m_t^2}{4\pi^2}$ | none (flat spacetime) |

The loop corrections (14)–(16) add up to form the quantum effective action

$$S(\eta, \psi, \Lambda_{\wp}) = S_{cl}(\eta, \psi) + \delta S_{\psi}(\eta, \log \Lambda_{\wp}) + \delta S_{\varnothing \phi}(\eta, \Lambda_{\wp}) + \delta S_V(\eta, \Lambda_{\wp}) \quad (17)$$

which represents long-wavelength sector of the exact QFT (namely the SM) in that all high-frequency quantum fluctuations have been integrated out to get the corrections (14)–(16). If this effective action does indeed act like classical field theories then it must get to curved spacetime as

$$S(g, \psi, \Lambda_{\wp}) \hookrightarrow S(g, \psi, \Lambda_{\wp}) + \int d^4x \sqrt{-g} \left\{ -\frac{\tilde{M}^2}{2} R(g) + \tilde{c}_2 R(g)^2 + \frac{\tilde{c}_3}{\tilde{M}^2} R(g)^3 + \dots \right\} \quad (18)$$

as follows from the equivalence principle map (10) and the curvature sector in (12). The curvature sector here is the one in (12). The problem with this curved spacetime action is that $\tilde{M}$, $\tilde{c}_2$, $\tilde{c}_3$, $\cdots$ are all bare constants carrying no loop corrections [29,31]. However, for a proper concord between the gravity and the SM all constants must be at the same loop level. This discord, which reveals the difference between truly classical and effective field theories, implies that effective QFTs do not allow put-by-hand curvature sectors; they necessitate curvature sector to be born from the effective action itself. In other words, curvature must arise from within $S(g, \psi, \Lambda_{\wp})$ itself so that no incalculable bare constants can arise in the curved spacetime action.

This result can be taken to imply that mass scales in the effective action $S(g, \psi, \Lambda_{\wp})$ must have some association with the spacetime curvature. To reveal the nature of this association and to determine if it is an equivalence relation it proves useful to give priority to the gauge boson mass action (14) as part of the SM effective action $S(\eta, \psi, \Lambda_{\wp})$ in (17).

It is clear that it breaks gauge symmetries explicitly, and continues to do so in the curved spacetime if carried there via the discordant map in (18).

Here comes to mind a critical question: Can one carry (14) into curved spacetime in a way restoring gauge symmetries? Can gauge invariance be the fundamental principle that governs how QFTs to be taken into curved spacetime? The answer will turn out to be affirmative. To see how, it proves efficacious to start with a simple identity [28,29,31]

$$\delta S_V\left(\eta, \Lambda_\wp^2\right) = \delta S_V\left(\eta, \Lambda_\wp^2\right) - I_V(\eta) + I_V(\eta) \tag{19}$$

in which the kinetic structure

$$I_V(\eta) = \int d^4x \sqrt{-\eta} \frac{c_V}{2} \mathrm{Tr}\left\{\eta_{\mu\alpha}\eta_{\nu\beta} V^{\mu\nu} V^{\alpha\beta}\right\} \tag{20}$$

is added to $\delta S_V\left(\eta, \Lambda_\wp^2\right)$ and subtracted back. Now, at the right-hand side of (19), if $\delta S_V$ is equated to (14), "$-I_V$" is kept unchanged and yet "$+I_V$" is expanded with by-parts integration, then, the identity (19) takes the form

$$\delta S_V\left(\eta, \Lambda_\wp^2\right) = -I_V(\eta) + \int d^4x \sqrt{-\eta} c_V \mathrm{Tr}\left\{V^\mu\left(-D^2_{\mu\nu} + \Lambda_\wp^2 \eta_{\mu\nu}\right)V^\nu + \partial_\mu\left(\eta_{\alpha\beta} V^\alpha V^{\beta\mu}\right)\right\} \tag{21}$$

where $V_{\mu\nu}$ is the field strength tensor of $V_\mu$, $D_\mu$ is the gauge-covariant derivative, with $D^2_{\mu\nu} = D^2 \eta_{\mu\nu} - D_\mu D_\nu - V_{\mu\nu}$ and $D^2 = \eta^{\mu\nu} D_\mu D_\nu$. This reshaped effective action becomes

$$\delta S_V\left(g, \Lambda_\wp^2\right) = -I_V(g) + \int d^4x \sqrt{-g} c_V \mathrm{Tr}\left\{V^\mu\left(-\mathcal{D}^2_{\mu\nu} + \Lambda_\wp^2 g_{\mu\nu}\right)V^\nu + \nabla_\mu\left(g_{\alpha\beta} V^\alpha V^{\beta\mu}\right)\right\} \tag{22}$$

under the equivalence principle map in (10) such that $\mathcal{D}_\mu$ is the gauge-covariant derivative in curved geometry, and $\mathcal{D}^2_{\mu\nu} = \mathcal{D}^2 g_{\mu\nu} - \mathcal{D}_\mu \mathcal{D}_\nu - V_{\mu\nu}$ with $\mathcal{D}^2 = g^{\mu\nu}\mathcal{D}_\mu\mathcal{D}_\nu$.

Now, a brief examination of (22) immediately reveals that it would vanish identically if $\Lambda_\wp^2 g_{\mu\nu}$ were replaced with $R_{\mu\nu}(^g\Gamma)$ since

$$\int d^4x \sqrt{-g} c_V \mathrm{Tr}\left\{V^\mu\left(-\mathcal{D}^2_{\mu\nu} + R_{\mu\nu}(^g\Gamma)\right)V^\nu + \nabla_\mu\left(g_{\alpha\beta} V^\alpha V^{\beta\mu}\right)\right\} = I_V(g) \tag{23}$$

as follows from by-parts integration [28,29,34]. This rather appealing feature is actually wholly flawed because $\Lambda_\wp^2 g_{\mu\nu} \hookrightarrow R_{\mu\nu}(^g\Gamma)$ contradicts with $\eta_{\mu\nu} \hookrightarrow g_{\mu\nu}$. If this contradiction were not there the CCB [49,50] would be solved by the metamorphosis of $\Lambda_\wp^2 g_{\mu\nu}$ into $R_{\mu\nu}(^g\Gamma)$. This contradiction can be prevented by implementing a more general map [34]

$$\Lambda_\wp^2 g_{\mu\nu} \hookrightarrow \mathbb{R}_{\mu\nu}(\Gamma) \tag{24}$$

by utilizing the Ricci curvature $\mathbb{R}_{\mu\nu}(\Gamma)$ of a symmetric affine connection $\Gamma^\lambda_{\mu\nu}$, which has nothing to do with the Levi–Civita connection $^g\Gamma^\lambda_{\mu\nu}$) [58–60]. The idea is that the equivalence principle that maps $\eta_{\mu\nu}$ to $g_{\mu\nu}$ is extended by the curvature map (24) to establish a relationship between $\Lambda_\wp^2 g_{\mu\nu}$ and $\mathbb{R}_{\mu\nu}(\Gamma)$. The contradiction is removed because the maps (10) and (24) involve independent dynamical variables [34]. The curvature map can also be interpreted as the substance needed to relate general covariance to gravity [42–44]. The curvature map (24) can also be interpreted as a Poincare affinity relation [28] in that $\Lambda_\wp^2$ (curvature) breaks the Poincare symmetry in flat spacetime (curved spacetime), and the two quantities assume a certain affinity that facilitates the curvature map in (24). As a result, the action (22) takes a completely new form

$$\delta S_V(g, \mathbb{R}) = -I(g, V) + \int d^4x \sqrt{-g} c_V \mathrm{Tr}\left\{V^\mu\left(-\mathcal{D}^2_{\mu\nu} + \mathbb{R}_{\mu\nu}(\Gamma)\right)V^\nu + \nabla_\mu\left(g_{\alpha\beta} V^\alpha V^{\beta\mu}\right)\right\} \tag{25}$$

under the maps (10) and (24), and simplifies to

$$\delta S_V(g, \mathbb{R}, R) = \int d^4 x \sqrt{-g}\, c_V \mathrm{Tr}\left\{ V^\mu \big( \mathbb{R}_{\mu\nu}(\Gamma) - R_{\mu\nu}(\,^g\Gamma) \big) V^\nu \right\} \tag{26}$$

by the identity (23). The gauge boson mass term in (14) has been transformed into a pure curvature term. The anomalous gauge boson masses have been completely metamorphosed but the resulting action, the action (26), is non-vanishing and gauge symmetries continue to be explicitly broken. This problem, the CCB [49,50], can have a solution only if $\mathbb{R}_{\mu\nu}(\Gamma)$ approaches to $R_{\mu\nu}(\,^g\Gamma)$, dynamically. The question of if and when this happens depends on the dynamics of the affine connection $\Gamma^\lambda_{\mu\nu}$.

With a view at determining the $\Gamma^\lambda_{\mu\nu}$ dynamics, it proves useful to start with the logarithmic UV sensitivities. The identity (23), which is what reduces the action (25) to (26), rests on the condition that $c_V$ must remain untouched under the curvature map (24). In other words, $\log \Lambda_\wp$, which can appear in $c_V$ at higher loops, must remain intact while $\Lambda_\wp^2$ transforms into affine curvature as in (24). This condition, which is in agreement with the Poincare-respecting (Poincare-breaking) nature of $\log \Lambda_\wp$ ($\Lambda_\wp^2$), puts the remnant QFT in dimensional regularization scheme [28,29,56,57]. Indeed, as follows from (16) via the maps (10) and (24), the logarithmic corrections in curved spacetime

$$\delta S(g, \psi, \log \Lambda_\wp) \supset -\sum_i \hat{c}_{\psi_i} m_i^4 \log \frac{m_i^2}{\Lambda_\wp^2} - \sum_{i,\phi} \hat{c}_{\phi\psi_i} m_i^2 \log \frac{m_i^2}{\Lambda_\wp^2} \phi^2 - \sum_{i,f} \hat{c}_{f\psi_i} \log \frac{m_i^2}{\Lambda_\wp^2} m_f \bar{f} f \tag{27}$$

lead to the log-regularized action

$$S_{\mathrm{QFT}}(g, \psi, \log \Lambda_\wp) = S_{cl}(g, \psi) + \delta S(g, \psi, \log \Lambda_\wp) \tag{28}$$

which can always be interpreted in the language of the dimensional regularization via the transformation [55–57]

$$\log \Lambda_\wp^2 = \frac{1}{\epsilon} - \gamma_E + 1 + \log 4\pi \mu^2 \tag{29}$$

which trades $\log \Lambda_\wp$ for $1/\epsilon$-substraction (MS) renormalization scale $\mu$ in $4 + \epsilon$ dimensions. In general, variations of the scattering amplitudes with $\mu$ are described by the renormalization group equations [55].

It is clear that, under the maps (10) and (24), the flat spacetime effective action (15) of the power-law UV corrections leads to the curvature sector

$$\text{``curvature sector''} = \int d^4 x \sqrt{-g} \left\{ -Q^{\mu\nu} \mathbb{R}_{\mu\nu}(\Gamma) + \frac{1}{16} c_\varnothing \big( g^{\mu\nu} \mathbb{R}_{\mu\nu}(\Gamma) \big)^2 - c_V R_{\mu\nu}(\,^g\Gamma) \mathrm{Tr}\{ V^\mu V^\nu \} \right\} \tag{30}$$

in which the disformal metric

$$Q_{\mu\nu} = \left( \frac{1}{4} \sum_i c_{\psi_i} m_i^2 + \frac{1}{4} c_\phi \phi^2 + \frac{1}{8} c_\varnothing g^{\alpha\beta} \mathbb{R}_{\alpha\beta}(\Gamma) \right) g_{\mu\nu} - c_V \mathrm{Tr}\{ V_\mu V_\nu \} \tag{31}$$

involves all the scalars $\phi$ and vectors $V_\mu$. This curvature sector satisfies the condition (5) in the Introduction.

The emergent curvature sector (30) takes the place of the by-hand curvature sector in (12). It removes the discord in (18) and induces the fundamental scale of gravity

$$M_{Pl}^2 = \frac{1}{2} \sum_i c_{\psi_i} m_i^2 \tag{32}$$

as a pure quantum effect. Its one-loop value $(M_{Pl}^2)_{1-\text{loop}} = \text{str}\{m^2\}/64\pi^2$, as follows from Table 1, ensures that, in the SM, $M_{Pl}^2$ comes out wrong in both size and sign. It cannot come out right unless there exists an additional matter in the spectrum because

$$\text{Planck scale necessitates a BSM sector} \tag{33}$$

whose bosons must either outweigh (some 1000 Planck-mass bosons) or outnumber (some $10^{32}$ electroweak-mass bosons) the fermions [28,29,31]. The other crucial feature is that:

$$\text{The BSM sector does not have to interact with the SM} \tag{34}$$

since $M_{Pl}^2$, supertrace of particle mass-squareds, is independent of if the SM + BSM fields are mixing or not [28,29,31]. This property is precisely what is needed to satisfy the condition (6) in the Introduction.

Stationarity of the action (30) against variations in $\Gamma_{\mu\nu}^\lambda$ leads to the $\Gamma_{\mu\nu}^\lambda$ equation of motion

$$^\Gamma\nabla_\lambda Q_{\mu\nu} = 0 \tag{35}$$

whose solution [58–62]

$$\Gamma_{\mu\nu}^\lambda = {}^g\Gamma_{\mu\nu}^\lambda + \frac{1}{2}(Q^{-1})^{\lambda\rho}\left(\nabla_\mu Q_{\nu\rho} + \nabla_\nu Q_{\rho\mu} - \nabla_\rho Q_{\mu\nu}\right) \tag{36}$$

remains in close proximity of the Levi–Civita connection ${}^g\Gamma_{\mu\nu}^\lambda$ thanks to the enormity of $M_{Pl}$. Corresponding to (36) the affine Ricci curvature expands as

$$\mathbb{R}_{\mu\nu}(\Gamma) = R_{\mu\nu}({}^g\Gamma) + \frac{1}{M_{Pl}^2}\left(\nabla^\alpha\nabla_\mu\delta_\nu^\beta + \nabla^\alpha\nabla_\nu\delta_\mu^\beta - \Box\delta_\mu^\alpha\delta_\nu^\beta - \nabla_\nu\nabla_\mu g^{\alpha\beta}\right)Q_{\alpha\beta} + \mathcal{O}\left(M_{Pl}^{-4}\right) \tag{37}$$

with a remainder which involves only derivatives of $\phi$ and $V_\mu$ [28,29].

The curvature solution (37) ensures that the CCB action in (26) gets strongly suppressed

$$\int d^4x\sqrt{-g}\sum_V c_V \text{Tr}\left\{V^\mu\left(\mathbb{R}_{\mu\nu}(\Gamma) - R_{\mu\nu}({}^g\Gamma)\right)V^\nu\right\} = \int d^4x\sqrt{-g}\left\{0 + \mathcal{O}\left(M_{Pl}^{-2}\right)\right\} \tag{38}$$

and thus the CCB gets prevented up to $\mathcal{O}\left(M_{Pl}^{-2}\right)$ effects [29]. Thus, the affine connection $\Gamma_{\mu\nu}^\lambda$ possesses right dynamics to suppress the problematic curvature difference (26).

The curvature solution (37) reduces the metric-affine curvature sector in (30) [58–63] to the GR curvature sector

$$\text{``curvature sector''} = \int d^4x\sqrt{-g}\left\{-\frac{1}{2}M_{Pl}^2 R(g) - \frac{1}{4}c_\phi\phi^2 R(g) - \frac{1}{16}c_\varnothing R(g)^2 + \mathcal{O}\left(M_{Pl}^{-2}\right)\right\} \tag{39}$$

which proves that the UV-sensitivity problems listed in Table 1 are all gone. The curvature scalar directly couples to scalar fields [64,65], with a quadratic part [66,67], which can lead to the Starobinsky inflation [68–70].

At long last, the SM + BSM and gravity come together concordantly

$$S_{\text{QFT}\cup\text{GR}} = S_{\text{QFT}}(g,\psi,\log\Lambda_\wp) + \int d^4x\sqrt{-g}\left\{-\frac{1}{2}M_{Pl}^2 R(g) - \frac{1}{4}c_\phi\phi^2 R(g) - \frac{1}{16}c_\varnothing R(g)^2 + \mathcal{O}\left(M_{Pl}^{-2}\right)\right\} \tag{40}$$

in which $M_{Pl}^2$, $c_\phi$ and $c_\varnothing$ are all bona fide loop-induced constants computed in the flat spacetime such that the remnant $\log\Lambda_\wp$ dependencies can always be put in the dimensional regularization language via the relation (29). These constants can be computed for a given SM + BSM, and the resulting curved spacetime effective SM + BSM of (40) can be tested via collider (such as the FCC and dark matter searches), astrophysical (such as neutron stars and black holes) or cosmological (such as inflation and structure formation) phenomena.

Increasing precision in collider experiments [71,72], astrophysical observations [9–11,73] and cosmological measurements [9–11,74,75] is expected to refine its BSM sector as well as the higher-curvature terms.

The whole mechanism above is termed as *symmergence* in that gravity emerges in a way restoring the gauge symmetries. Symmergence differs from the known completions of the SM by its ability to satisfy the conditions (5) and (6), and predict the existence of a BSM sector. For clarification, it proves useful to contrast symmergence with another method based on subtraction of the $\Lambda_\wp^2$ terms [76]. This method is particularly relevant in that it subtracts (absorbs into critical surface) $\Lambda_\wp^2$ terms, retains only $\log \Lambda_\wp$ terms (dimensional regularization), predicts no BSM sector and leaves out gravity entirely. Symmergence [28,29,31], on the other hand, sets an equivalence between $\Lambda_\wp^2$ and spacetime curvature via their Poincare affinity, retains only $\log \Lambda_\wp$ terms (dimensional regularization), predicts the existence of a BSM sector and makes gravity emerge in a way restoring the gauge symmetries.

## 3. Naturally Coupled Dark Sector

Symmergence has reconciled gravity with the SM + BSM at the same loop level. It has solved the problems in Table 1. This, however, is only the $\Lambda_\wp^2$ part of the Higgs mass correction in (4). Its logarithmic part remains as a potential source of hierarchy problem for heavy BSM.

The $\mathcal{O}\left(M_{Pl}^{-2}\right)$ remainder in (40), which results from the higher-orders of the curvature expansion in (37), implies that the scalar and gauge fields in SM + BSM can have doubly Planck-suppressed, non-renormalizable, all-derivative interactions. These tiny interactions can hardly have any significant impact on low-energy processes. In what follows, therefore, analyses of the DS-SM couplings will be limited to renormalizable-level interactions between the two.

The problem, as revealed by the logarithmic action in (27), is that the heavy BSM fields $H', V', N', \ldots$ pull the Higgs boson mass (and the vacuum energy) towards $M_{BSM}$ via their various couplings (similar to the portals in (1)–(3)). This SM-BSM hierarchy problem (the little hierarchy problem [77–79] of supersymmetry) is as important as the UV-induced ones listed in Table 1. In fact, known UV completions of the SM [7,8] (supersymmetry, extra dimensions, compositeness and their hybrids) have been sidelined by the LHC experiments due to this SM-BSM hierarchy problem in that their BSM sectors (superpartners in supersymmetry, Kaluza–Klein levels in extra dimensions and technifermions in compositeness) invariably require $\lambda_{SM-BSM} \simeq \lambda_{SM}$ by their symmetry structures ($\lambda_{SM}$ is a typical SM coupling).

Symmergence is not sidelined by the LHC experiments. It survives the bounds. The reason is that it predicts the existence of a BSM sector (as derived in (33)) that does not have to couple to the SM (as found in (34)). The SM and BSM can have no coupling ($\lambda_{SM-BSM} = 0$), weak coupling ($\lambda_{SM-BSM} \ll \lambda_{SM}$), or sizable coupling ($\lambda_{SM-BSM} \simeq \lambda_{SM}$) as long as the experimental and observational constraints are satisfied. This freedom is a feature of symmergence and, as will be discussed below, the BSM is essentially a dark sector since $\lambda_{SM-BSM} \simeq \lambda_{SM}$ is already disfavored by the LHC. To clarify further one notes that in supersymmetry, extra dimensions, compositeness and their hybrids [7,8]:

$$\lambda_{SM-BSM} \simeq \lambda_{SM} \implies \text{BSM must be light! } (M_{BSM} \gtrsim m_h) \tag{41}$$

whereas in symmergence [29,31]:

$$\lambda_{SM-BSM} = \text{``small''} \implies \text{BSM can lie at any scale! } (M_{BSM} \gtrsim m_h, M_{BSM} \gg m_h) \tag{42}$$

so that the LHC experiments are able to sideline sparticles (the BSM of supersymmetry), Kaluza–Klein levels (the BSM of extra dimensions) and technifermions (the BSM of com-

positeness) but not the BSM of the symmergence! In fact, if the SM-BSM couplings obey the bound

$$\lambda_{SM-BSM} \lesssim \frac{m_H^2}{M_{BSM}^2} \tag{43}$$

then the correction (4) to Higgs boson mass resides within the LHC bounds. This bound imposes a seesawic relation between the Higgs condensation parameter $m_H^2$ ($m_h^2 = -2m_H^2$ in the SM) and the BSM masses. This seesawic relation has important implications for collider and other searches: It means heavier the BSM smaller its coupling to the SM (the SM is decoupled from heavy sectors). It means also heavier the BSM larger the luminosity needed to probe it at colliders. It further means heavier the dark matter (belonging to the BSM) smaller its scattering cross section from the nucleons and harder its detection in direct searches. In this sense, seesawic coupling has novel implications for collider and other searches.

The BSM whose coupling to the SM obey the bound in (43) will hereon be called *naturally coupled BSM* because fields of such BSM do not destabilize the SM Higgs sector. In a naturally coupled BSM, where BSM = VS + DS, the two subsectors possess the following features:

1.  The **VS** is endowed with SM charges and thus $\lambda_{SM-VS}$ is fixated to be $\lambda_{SM-VS} \simeq \lambda_{SM}$ as in (41). This means that $M_{VS} \gtrsim m_h$ is a necessity. In other words, the VS sector must weigh near the weak scale. This VS structure is that of the supersymmetry and other completions, and is not favored by the current LHC searches.
2.  The **DS** is SM-singlet as a whole and thus $\lambda_{SM-DS}$ is free to take any perturbative value. It has the form in (42). This means that $M_{DS}$ can lie at any scale from $m_h$ way up to ultra-high scales.

These properties ensure that the naturally coupled heavy BSM is essentially *naturally coupled DS*, and the naturalness bound in (43) is effective mainly for the DS. In this regard, the DS is composed of two kinds of particles:

1.  **Black Particles.** These kinds of particles have zero non-gravitational couplings to the SM [80,81]. They form a secluded sector formed, for instance, by high-rank non-Abelian gauge fields and fermions [31,35,82,83]. In view of the present searches, which seem all negative, this black (pitch-dark) DS agrees with all the available data [84,85]. It is unique to symmergence in that $\lambda_{SM-DS}$ obeys the naturalness bound in (43) and hence can well be vanishing, $\lambda_{SM-Black\ DS} = 0$. In this case, the DS possesses only gravitational couplings to the SM.
2.  **Dark Particles.** This type of particles are neutral under the SM gauge group and couple to the SM via the Higgs portal in (1), hypercharge portal in (2) and the neutrino portal in (3) [9–11,31]. Indeed, scalars $H'$, Abelian vectors $V'_\mu$ and fermions $N'$ are dark particles and couple to the SM at the renormalizable level via

$$S_{int} = \int d^4x \sqrt{-g} \left\{ \lambda_{H'}^2 \left( H^\dagger H \right) \left( H'^\dagger H' \right) + \lambda_{Z'} B_{\mu\nu} Z'^{\mu\nu} + \left[ \lambda_{N'} \overline{L} H N' + \text{h.c.} \right] \right\} \tag{44}$$

in which the couplings $\lambda_{H'}^2$, $\lambda_{Z'}$, $\lambda_{N'}$ can take, from none to significant, a wide range of values with characteristic experimental signals [29,31,84,85].

The DS, whose two subsectors are depicted in Figure 1, can have any kind of particles, can have any perturbative couplings and can lie at any scale (from the weak scale to way up to the Planck scale) as long as the gravitational scale in (32) comes out right. Non-necessity of any sizable DS-SM couplings, as derived in (34), is what distinguishes the DS of the symmergence from the BSM sectors of supersymmetry, extra dimensions, composite models and others [7,8] where a sizable SM-BSM coupling is a symmetry requirement. The schematic plot in Figure 2 is the variation of the DS-SM coupling with the BSM masses $M_{F'}$. The region allowed by natural couplings (the bound in (43)) is the one below the curve.

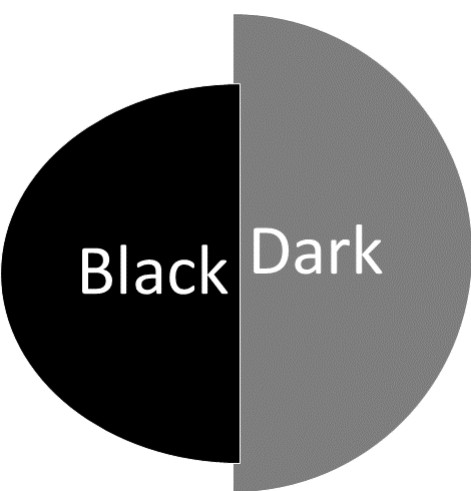

**Figure 1.** The DS of symmergence has two distinct sectors: The black particles which have only gravitational couplings to the SM, and dark particles that have gravitational couplings as well as the portal couplings in (44). The current collider and direct search bounds do actually agree with a black sector.

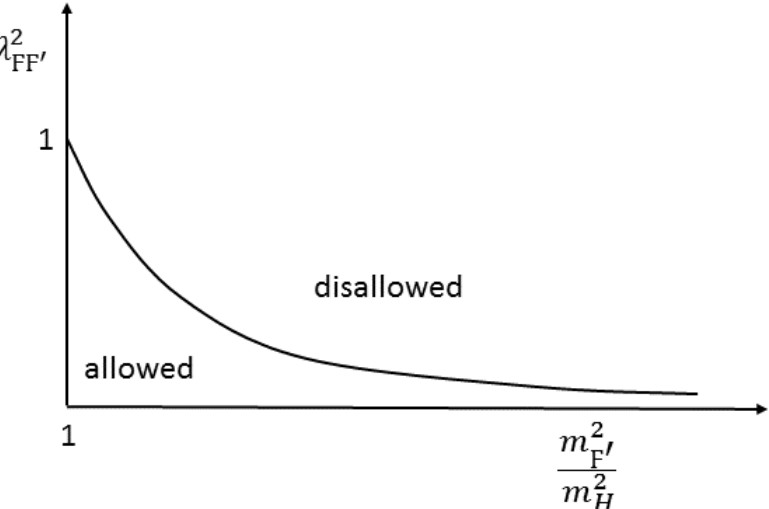

**Figure 2.** The allowed and disallowed regions for a naturally coupled DS obeying the bound in (43). Here $F'$ is a typical DS field mentioned in (1)–(3). (taken from Reference [29]).

To elucidate the meaning and effects of the naturally coupled DS; it proves convenient to study its certain salient features. The first point to study would be naturalness condition (43) under the renormalization group flow of the model parameters. This is important because the Higgs boson mass mast be kept stable under the logarithmic contributions in (44). Depicted in Figure 3 are the scale dependencies of the Higgs boson mass from the electroweak scale up to the GUT scale in the SM (black curve) and in the SM + DS (red curve). In here, the DS is represented by a singlet scalar $H'$ such that SM–DS coupling obeys the seesawic bound in (43) at the GUT scale. It thus follows that naturally coupled dark sectors do indeed not disrupt the stability of the electroweak scale [38].

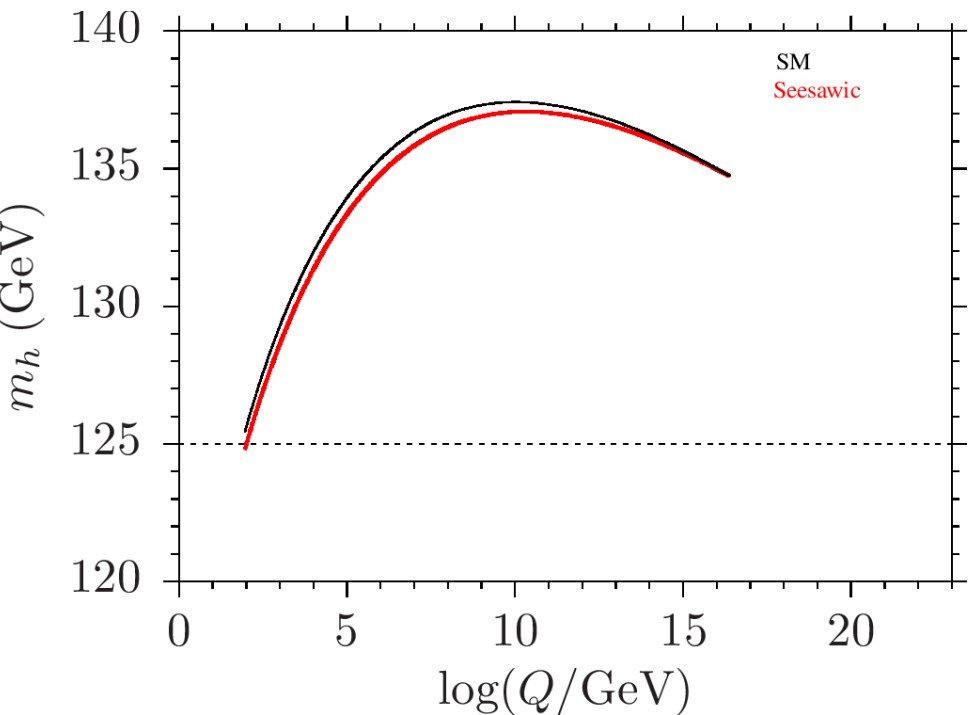

**Figure 3.** The dependence of the Higgs boson mass on the mass scale $Q$ in the SM (black curve) and in the SM + DS with DS containing a scalar $H'$ coupling to the SM as in (44) obeying the naturalness bound in (43). The naturally coupled DS does indeed respect the stability of the electroweak scale. (taken from Reference [38], with the permission of the co-author.)

The Higgs boson scatterings can probe a naturally coupled DS. What needs be done is to measure the scalar and vector masses and then determine if their production cross sections are in accord with the seesawic couplings in (43). For instance, $m_{H'}$ can be extracted from $HH \rightarrow H' \rightarrow HH$ in Figure 4a or $HH \rightarrow H'H'$ in Figure 4b. Then the ratio of their cross sections

$$\frac{\sigma(HH \rightarrow H' \rightarrow HH)}{\sigma(HH \rightarrow H'H')} \simeq \left(\frac{m_H^2}{m_{H'}^2}\right)^2 \tag{45}$$

acts as a decisive probe of a naturally coupled heavy scalar $H'$ in the DS. This method works also for the $Z'$ scatterings in Figure 4c,d. The scatterings as such (specifically the $2 \rightarrow 2$ scatterings here) can be directly tested at present [86,87] and future [88,89] colliders, and decisions can be made on the seesawic nature of the DS [39].

To refine the the approximate analysis above, production of $H'$ at the LHC can be studied via its mixing with the SM Higgs field as in (44) under the naturalness bound in (43). In this regard, production cross sections for a set of final states are plotted in Figure 5 in which $h_2$ is the heavy mix of the $H$ and $H'$. It is clear that cross sections fall with $M_{H'}$ in accordance with the naturalness bound in (43). It is with higher and higher luminosities that a collider can access heavier and heavier particles [39].

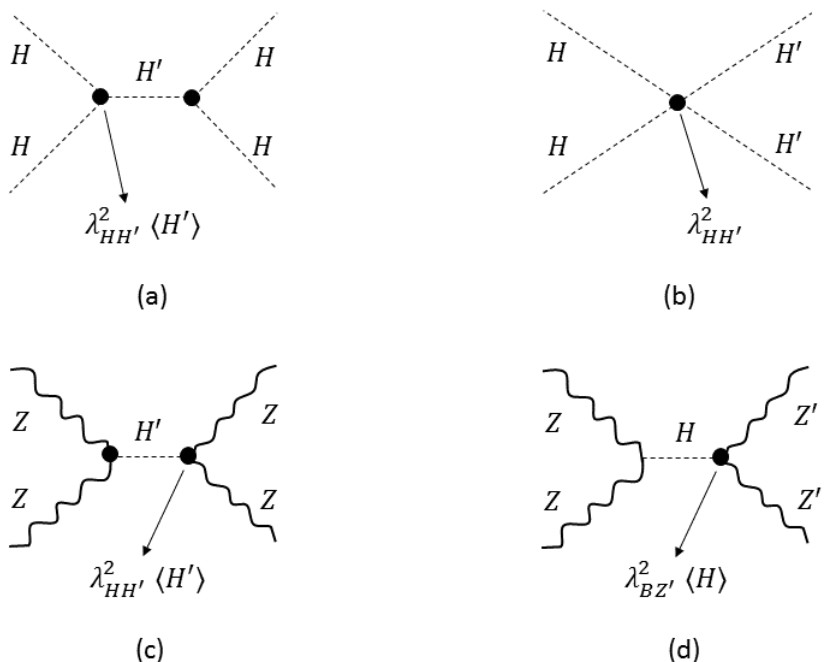

**Figure 4.** Basic search channels for the BSM scalars $H'$ (**a,b**) and BSM vectors $V'_\mu$ (**c,d**). (taken from Reference [29]).

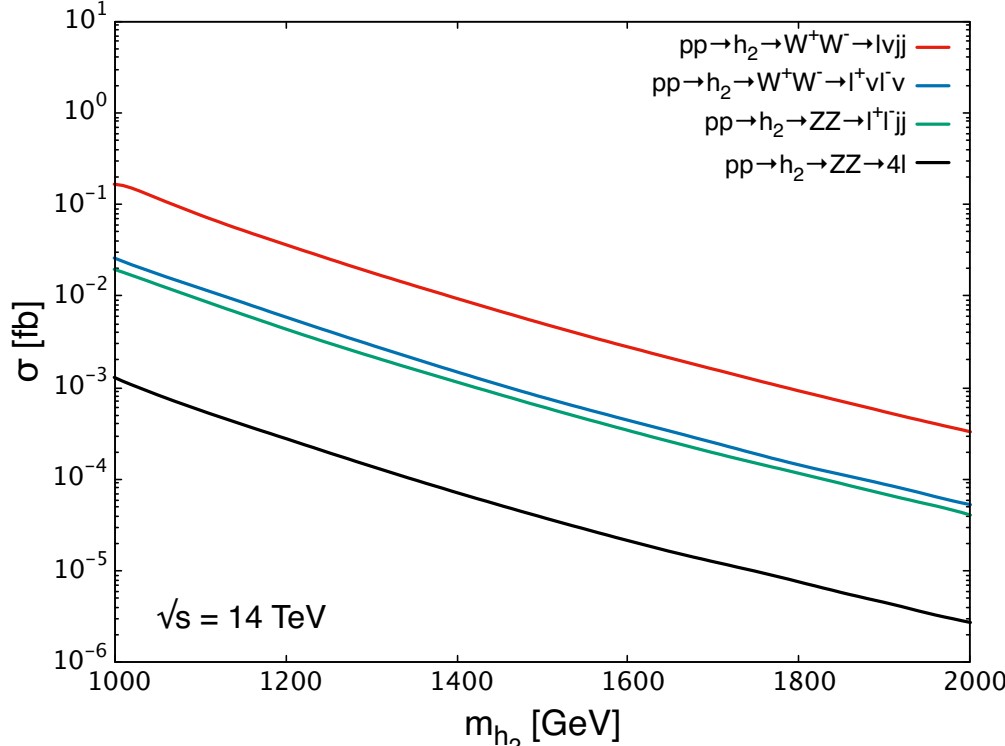

**Figure 5.** Effects of the heavy DS scalar $H'$ on the production of jets and leptons. The cross sections fall with the $H'$ mass in accordance with the naturalness bound in (43). In general, the heavier the naturally coupled dark sectors, the larger the luminosity needed to detect them. The negative searches at the LHC may be a result of the naturally coupled nature of the dark sector scalars. (taken from Reference [39], with the permission of the co-authors).

The naturally coupled dark matter is an important particle to study. Symmergence has candidates for dark matter in both the black and dark DS sectors. The black matter satisfies

all the bounds at present. It is with direct experiments or observations that one can determine if the dark matter is black or dark (such as, for instance, less-baryonic galaxies similar to cosmic seagull [90]). The DS scalar in (44) can well be a dark matter candidate, and it can be probed directly at the LHC [91] and direct searches [84,85]. With a strictly vanishing vacuum expectation value $\langle H' \rangle = 0$ (with an odd $Z_2$ partity with respect to the Higgs field) $H'$ possesses no decay channels and gains absolute stability. Then its number density gets depleted due to expansion of the Universe as well as its coannihilations into the SM particles via the processes $H'H' \rightarrow W^+W^-, ZZ, \bar{t}t, \cdots$. With the scattering in $H'H' \rightarrow HH$ in Figure 4b (equivalent to annihilations into $W/Z$ by the Goldstone equivalence theorem), the $H'$ scalars attain the observed relic density [84,85,92,93] if

$$\lambda_{HH'}^4 \left( \frac{\text{GeV}}{m_{H'}} \right)^2 \approx 10^{-7} \tag{46}$$

which implies $M_{H'} \leq 400\,\text{GeV}$ under the natural couplings in (43) for $\lambda_{HH'}$. This bound on $m_{H'}$ implies that thermal dark matter, if saturated by a single scalar field such as $H'$, must not weigh above the electroweak scale. This means that, under mild assumptions about the structure of the dark matter, naturally coupled DS can be probed by measuring the dark matter mass [84,85,91].

In addition to the approximate analysis above, the dark matter search can be analyzed by detailed simulations with the imposition of the natural couplings in (43). Indeed, for a scalar dark matter candidate $H'$, which couples to the SM Higgs fields as in (44), the spin-independent scattering cross section from the nucleons vary with $M_{H'}$ as in Figure 6. The scan of the parameter space gives a relatively wide region (green) but the naturalness condition in (43) restricts allowed range to the blue strip. The blue cross section is seen to fall with $m'_H$ in accordance with (43) according to which heavier the $H'$ smaller its cross section. The negative results at dark matter direct searches may thus be an indication of the naturally coupled nature of the dark matter [38].

The right-handed neutrinos are the only known particles beyond the SM. If they couple as in (3) with natural couplings in (43) then they are constrained to be relatively light

$$m_{N'} \lesssim 1000\,\text{TeV} \tag{47}$$

for active neutrinos to be able to acquire experimentally admissible Majorana masses [94–96]. The naturally coupled right-handed neutrinos might be probed directly at future experiments (combining Higgs factories and accelerator neutrinos [97]) if not indirectly at the near-future SHiP experiment [98,99].

In this section, focus has been on the types of dark sectors, their natural couplings to the SM, and salient ways of probing them. To better see what novelty the symmergence brings up, it proves convenient to compare the analyses above with similar ones in the literature. Indeed, the SM ports to dark sector haven been studied in great detail in the literature (see the review volumes [23] for the Higgs portal, [24] for the hypercharge portal and [100] for the neutrino portal). In these studies, the portal couplings are subjected to collider bounds (the LHC and others) on one side and astrophysical and cosmological bounds (such as dark matter searches) on the other. The model space is scanned with dedicated simulation codes to determine allowed ranges of various couplings. In doing so, however, loop corrections to the SM parameters (the Higgs boson mass, for instance) are seldom taken into account (except for supersymmetric constructions, for instance). In other words, the electroweak stability constraints (5) nor (6) are seldom taken into consideration. The point is that generic portal models (with additional fields from the dark sector) can in general not satisfy both of the constraints (5) nor (6). The cutoff (or matching) scale $\Lambda_\wp$, lying above the BSM scale $M_{BSM}$, is there as a physical scale indicative of gravity if not other possible high-scale interactions. If the constraint (6) is imposed than the scanned parameter space shrinks to that of the natural couplings (43). (This feature is illustrated by the wide green (general couplings) and narrow blue (natural couplings) regions in

Figure 6.) However, then the constraint (5) remains unneutralized and the Higgs boson mass jumps to $\Lambda_\wp$. Alternatively, if one imposes the constraint (5) first (by invoking a supersymmetric structure, for instance) then the constraint (6) remains unneutralized (since symmetry of the model does not allow SM–DS couplings to take small values). In the end, the Higgs boson mass jumps to $M_{BSM}$. The conclusion is that the generic portal couplings typically destabilize the electroweak scale, and various phenomenological discussions in the literature seldom discuss this problem (see for instance the supersymmetry section of the review [23]). It is here that symmergence stands out as a model proper to resolve both of the constraints (5) and (6). It is in this sense that naturally coupled dark sectors are a feature of the symmergence, and their revelation by experiments or observations will point towards a mechanism such as symmergence.

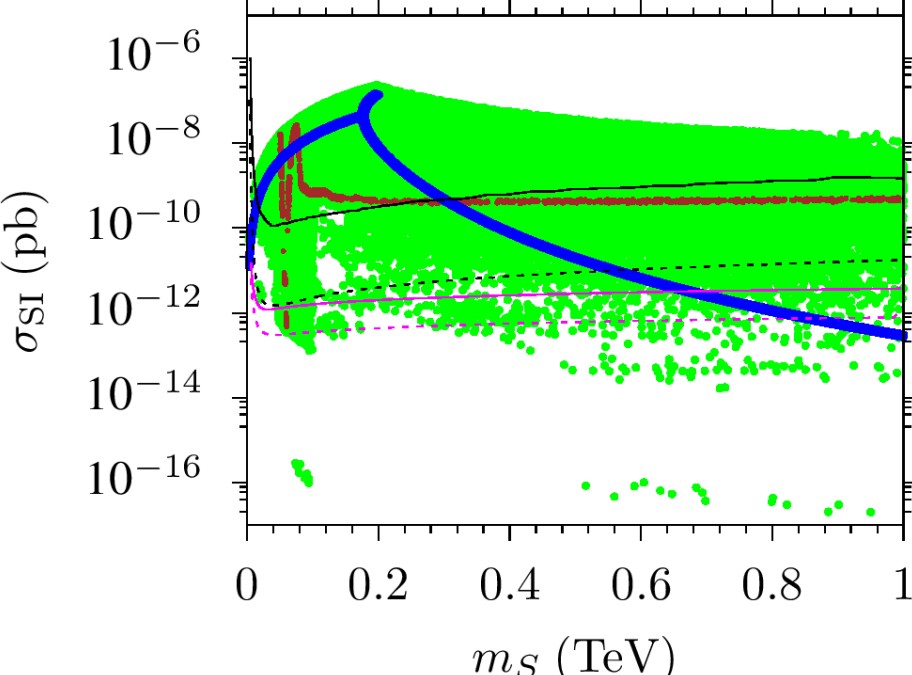

**Figure 6.** Variation of the dark matter–nucleon spin-independent cross section with the dark matter mass ($m_S$ is the same as $M_{H'}$). The scan of the parameter space gives a relatively wide region (green) but the naturalness condition in (43) restricts allowed range to the blue domain. The blue cross section is seen to fall with $m'_H$ in accordance with (43) according to which the heavier the $H'$ the smaller its cross section. The negative results at dark matter direct searches may be an indication of the naturally coupled nature of the dark matter. (taken from [38], with the permission of the co-author).

## 4. Conclusions and Future Prospects

In this mini review, the two physical requirements, References (5) and (6), which are what are expected of a natural completion of the SM, have been discussed in depth, and symmergence has been shown to satisfy both of them. Indeed, symmergence, as studied in Section 2, predicted the existence of a BSM sector (33), guaranteed that the SM and BSM do not have to interact (34), and incorporated gravity into SM + BSM and solved this way the notorious UV-sensitivity problems of the SM. More specifically, symmergence has led to the effective action $S_{QFT\cup GR}$ as an intertwined whole of the SM + BSM and gravity, where the BSM and gravity sectors emerge via the symmergence. Its gravitational and field-theoretic parts are at the same loop level with mere $\log \Lambda_\wp$ dependence (dimensional regularization). Unlike the known UV completions, which cannot satisfy both of the constraints (5) and (6), symmergence satisfies both of the constraints with allowance to natural couplings in (43) with no fine-tuning in SM–DS interactions. It is in this sense that naturally coupled

dark sectors can be taken to be an indicative of the symmergence in collider as well as astrophysical searches.

We have revealed possibility of naturally coupled dark sectors in symmergence, characterized their spectra (black and dark fields) and exemplified salient processes that can help probing them. The processes include collider experiments, dark matter searches, neutrino masses and possibly also dark energy models. The sample studies in this review can be extended to multi-field, multi-scale processes involving BSM particles such as flavons, axions, Affleck–Dine baryogenesis and natural inflation. The naturally coupled dark sectors become relevant also in models of dark energy. It can be the cosmological constant (coming from the logarithmic part (16)) or can be a dynamical field as has been modeled variously [9–11]. The latter must couple to the SM naturally via the seesawic couplings in (43) if the electroweak scale is to remain not destabilized. The naturally coupled dark sectors, whichever phenomena they appear as, can be regarded as an indicative of the symmergence as they can hardly arise in other completions of the SM. In this sense, the various phenomena discussed above can reveal distinctive signatures of symmergence in collider processes as well as astrophysical and cosmological phenomena.

This mini review is about the naturally coupled dark sectors, which are a special prediction of the symmergence. However, effects of symmergence are not limited to the naturally coupled dark sectors (though some of them can be viewed in this class). To this end, for the sake of completeness, it proves useful to mention (with no analysis) some of its important effects. First, the quadratic curvature term in (40), proportional to $c_\varnothing$, leads to the Starobinsky inflation [70]. The inflaton occurs in the Einstein frame as an SM-singlet scalar field after a conformal transformation of (40).

Second, the non-minimal coupling in (40), proportional to $c_\phi$, may realize multi-field plateau inflation with the inclusion of a number of scalar fields [101,102].

Third, the symmergent effective action (40) provides a framework for studying decays, scatterings and productions of particles in curved spacetime. The action (40) is the generator of one-particle-irreducible diagrams (vertices), which encode essential physics with all masses and couplings computed at the desired loop level. Their connections with propagators lead to connected diagrams, and replacement of the external legs in the connected diagrams with appropriate particle and anti-particle wavefunctions result in the *S*-matrix elements [49]. The *S*-matrix can be regarded as that of a semi-classical field theory in that in the effective action (40) all the fields are long-wavelength mean fields. In fact, as already illustrated in [28,40], scattering amplitudes can be thought of as containing wavefunctions in the relativistic quantum mechanics. This means that the decays and scatterings of particles can be approximated with those in the classical field theories. Then, as a future prospect, in view of its nearly classical structure, the symmergent setup in (40) can be utilized to study particle creation by gravitational field, Hawking radiation from black holes, semi-classical Einstein field equations (vacuum expectation value of the energy–momentum tensor) and quantum fluctuations in the early Universe. A detailed study of these phenomena is warranted because their analyses are hampered by fundamental difficulties pertaining to QFTs in curved spacetime. Indeed, in curved spacetime QFTs the requisite wavefunctions become ambiguous due to the fact that general covariance does not allow a unique vacuum state [103–105]. In fact, it is for this reason that particles and their scatterings become tractable mainly in asymptotically flat or conformally flat geometries [106–108] in which one makes use of the flat spacetime wavefunctions. It thus is clear that a detailed study of the symmergent gravity in strong gravity domains can reveal novel effects not found in the known approaches.

In view of these points, especially the third point, it is clear that symmergence provides a framework in which various astrophysical and cosmological phenomena can be studied. In addition, depending on the details of the dark sector, symmergence offers a rich collider and dark matter phenomenology.

**Funding:** This research was funded by TÜBİTAK grant 118F387.

**Institutional Review Board Statement:** Not applicable.

**Informed Consent Statement:** Not applicable.

**Acknowledgments:** The author is grateful to H. Azri, K. Cankoçak, C. Karahan, C. S. Ün, O. Sargın and S. Şen for collaborations on various parts of this review volume.

**Conflicts of Interest:** The authors declare no conflict of interest.

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
