# Peer review of "Naturally-Coupled Dark Sectors"

_galaxies, doi:10.3390/galaxies9020033_

Round 1
Reviewer 1 Report
This mini-review paper presents a study of the so-called dark sector. This subject is very popular now. In fact, dark energy and dark matter challenge the fundamental science. The author tries to construct an original approach in order to derive the dark sector from matching of quantum field theory (QFT) with gravity. The review combines several earlier published results. But there are severe problems in the very logics of such a construction. First of all, the author ignores the well known fact that QFT in general and the Standard Model in particular are well defined in curved space-times, see e.g. the textbook "Quantum Fields in Curved Space" by N. D. Birrell and P. C. W. Davies. And contrary to what is stated in the Introduction, the Standard Model doesn't need any TeV scale cutoff since it is a renormalizable theory. The real problem in matching of SM with gravity is in the absence of quantization of the latter. The author ignores this problem even when discusses the Plank energy scale. The very derivation of the BSM sector is based on a unjustified assumption that it should have renormalizable interactions with SM particles. Such an assumption is typically broken in effective models. As concerns the presented studies on "black" and "dark" particles, they do not add practically nothing to the field, since the Higgs, neutrino, and hypercharge portals have been already systematically studied in the literature at a much more advanced level.
Author Response
I thank to Reviewer 1 for his/her criticism and suggestions. I have gone through her/his comments, and made necessary improvements in the text. Below, I respond to the comments by the reviewer, and indicate changes in the paper:
A) First of all, the author ignores the well known fact that QFT in general and the Standard Model in particular are well defined in curved space-times, see e.g. the textbook "Quantum Fields in Curved Space" by N. D. Birrell and P. C. W. Davies.
Response: In principle, formally, one can define QFTs in curved spacetime. But, in classical curved spacetime, general covariance does not allow for a special state like the vacuum state, and absence of a unique vacuum makes the notion of particle ambiguous. It is in this sense that particle scatterings become tractable in asymptotically-flat or conformally-flat geometries. In symmergence, however, the resulting setup is an effective field theory setup, and particle scatterings can be safely discussed in the semi-classical with loop-improved masses and couplings. [I the text, I gave a detailed discussion of these points in Conclusion and Future Prospects section from line 439 to 461. I added references Ref. [72] and Ref. [74].]
B) And contrary to what is stated in the Introduction, the Standard Model doesn't need any TeV scale cutoff since it is a renormalizable theory.
Response: The SM is a renormalizable field theory by itself. It is all good in isolation. But the new physics beyond it (gravity and some extra QFT which encodes dark matter, dark energy, axion etc.) can be assumed to arise at some UV scale, and that scale turns out to be a TeV withe SM particle spectrum (Ref. [3]). [In the text, I extended the first paragraph of Introduction (from line 13 to 17) to explain this point.]
C) The real problem in matching of SM with gravity is in the absence of quantization of the latter. The author ignores this problem even when discusses the Plank energy scale.
Response: I thank to the referee for pointing out this important point. I included this point in the revised version.
[In the text, I revised Sec. 2 from line 74 up to line 101, and mentioned this important point from line 90 to 95. I added Ref. [30].]
D) The very derivation of the BSM sector is based on a unjustified assumption that it should have renormalizable interactions with SM particles. Such an assumption is typically broken in effective models.
Response: Yes, the SM-BSM interactions need not be renormalizable. I chose to emphasize renormalizable-level portal interactions. But I had to explain also non-renormalizable interactions (doubly-Planck suppressed interactions occurring in symmergence).
[In the text, I mentioned this point from the line 21 to 23 in the Introduction, and from the line 240 to 245 in Sec. 3.]
E) As concerns the presented studies on "black" and "dark" particles, they do not add practically nothing to the field, since the Higgs, neutrino, and hypercharge portals have been already systematically studied in the literature at a much more advanced level.
Response: It is true that the Higgs, neutrino and hypercharge portals have been analyzed in great detail. Those analyses do, however, seldom deal with loop corrections to the Higgs boson (and other boson) masses. They mainly scan the parameter spaces to determine the allowed ranges of the portal couplings, with no emphasis on the electroweak stability. In fact, the black matter satisfies all the existing bounds on dark sector. Likewise, naturally-coupled dark sectors maintain electroweak stability, and lead observable effects at collider experiments and astrophysical phenomena. In fact, imposition of electroweak stability (natural couplings) reduce the parameter space to that of symmergence, with no solution for power-law corrections. In this sense, symmergence provides a consistent framework in which natural portion of the allowed parameter space is identified, as in Fig. 6.
[In the text, I mentioned these points from line 371 to 399 in Sec. 3. I also added Ref. [67]. ]
I must say that I also reworded Abstract to better reflect content of the review.
Reviewer 2 Report
The author reviewed 'Symmergence', a mechanism of incorporating gravity into the Standard Model (SM) that allows a dark sector to be naturally coupled to the SM. The main difference between the proposed mechanism and other known completion of SM is that the SM-BSM couplings can be sufficiently small, which allows Symmergence to survive the LHC constraints. In this scheme, the higher the BSM scale is, the smaller the SM-BSM couplings.
The paper is a review based on several published papers. The discussion of the theory part is reasonably clear and complete. It is probably useful to have a review paper to put them together.
However, I am concerned that the discussion of "distinctive signatures which can be probed via cosmological, astrophysical and collider processes" is quite sketchy. One prediction is Eq. 43, which can be probed in LHC or future colliders. But it is more a test of the seesawic relation Eq. 41, rather than Symmergence itself directly. Similarly, a scan of the dark matter-nucleon spin-independent cross section vs. dark matter particle mass (Fig. 6) only gives a test of Eq. 41. The author did not really discuss cosmological or astrophysical signatures in any detail at all.
The author should consider extending the discussion of experimental/observational tests of Symmergence, particularly on what distinctive signatures there are to support Symmergence.
Author Response
I thank to Reviewer 2 for her/his comments and suggestions. I have gone through her/his comments, and made necessary improvements in the article.
A) However, I am concerned that the discussion of "distinctive signatures which can be probed via cosmological, astrophysical and collider processes" is quite sketchy. One prediction is Eq. 43, which can be probed in LHC or future colliders. But it is more a test of the seesawic relation Eq. 41, rather than Symmergence itself directly. Similarly, a scan of the dark matter-nucleon spin-independent cross section vs. dark matter particle mass (Fig. 6) only gives a test of Eq. 41. The author did not really discuss cosmological or astrophysical signatures in any detail at all.
Response: I think discussions in the original version were somewhat disorganized and vague, and this has misrepresented the content of the paper. Actually, the natural couplings (or seesawic couplings) themselves are a distinctive feature or an indicator of the symmergence. The reason is that it provides a framework in which loops corrections growing with the BSM scale and with the UV cutoff above it are both suppressed or neutralized. This is not the case in other known completions of the SM.
[In the text, I improved Introduction with particular care to Eqs. (5) and (6). I then added a discussion of the symmergence from the line 371 to 399 in Sec. 3 (I hope these changes answer Reviewer 2's question: "One prediction is Eq. 43, which can be probed in LHC or future colliders. But it is more a test of the seesawic relation Eq. 41, rather than Symmergence itself directly." properly.) I also extended the Conclusion and Future Prospects section from line 406 to 414 again to explain the essence of symmergence in comparison to existing analyses (parameter scans) in the literature on the portal couplings. I added Ref. [67].]
B) The author should consider extending the discussion of experimental/observational tests of Symmergence, particularly on what distinctive signatures there are to support Symmergence.
Response: Yes, I had to give more signatures to reveal the novelty brought by symmergence. In this revised version, I gave a discussion of the salient features in the symmergence in regard to curved spacetime QFT, mainly.
[In the text, I mentioned further distinctive aspects of symmergence from
line 430 to 465 in Conclusion and Future Prospects section. I added Refs. [68,69,72,74].]
I must say that I also reworded Abstract to better reflect content of the review.
Reviewer 3 Report
This review is well articulated and summarises several reference works. I believe that for publication it is necessary to extend the conclusions section, summarizing: i) what are the distinctive signatures of the naturally-coupled dark sector that can be seen from cosmological, astrophysical and collider processes data; ii) what is the observational evidence that supports the study of such models.
Author Response
I thank to Reviwer 3 for her/his comments and suggestions. I have gone through her/his comments, and made necessary improvements in the text.
Below, I respond her/his comments one by one:
i) what are the distinctive signatures of the naturally-coupled dark sector that can be seen from cosmological, astrophysical and collider processes data;
Response: After a rereading the original manuscript in view of Reviewer 3's report I came to realize that the discussions in the original manuscript were somewhat disorganized and vague. In reality,
the natural couplings (or seesawic couplings) themselves (along with the neutralization of the power-law-in-UV scale corrections) are a distinctive feature or an indicator of the symmergence. The reason is that it provides a framework in which loops corrections growing with the BSM scale and with the UV cutoff above it are both suppressed or neutralized. This is something specific to symmergence, and signatures like scalar dark matter or scalar field scatterings probe symmergence by probing the natural couplings.
[In the text, I improved Introduction. I added a discussion of the symmergence from the line 371 to 399 in Sec. 3. I also extended the Conclusion and Future Prospects section from line 406 to 414 again to explain symmergence in comparison to existing analyses (parameter scans) in the literature on the portal couplings. I added Ref. [67].]
ii) what is the observational evidence that supports the study of such models.
Response: As a I mentioned in i) the very confirmation of the seesawic couplings would be a confirmation of the symmergence mechanism. But symmergence has much more than that. I this regard, I gave a discussion of the salient features of symmergence in regard to curved spacetime QFT and inflation.
[In the text, I discussed distinctive aspects of symmergence from
line 430 to 465 in Conclusion and Future Prospects section. I added Refs. [68,69,72,74].]
I must say that I also reworded Abstract to better reflect the content of the review.
Round 2
Reviewer 1 Report
The author replied to all remarks contained in my first report. The corresponding changes are made in the text. I don't agree with all statements in it, but that can be retained as a basis for a scientific discussion. So, I suggest publishing the article as it is.